# Marine Microbial Polysaccharides: An Untapped Resource for Biotechnological Applications

**DOI:** 10.3390/md21070420

**Published:** 2023-07-24

**Authors:** Rajesh Jeewon, Aadil Ahmad Aullybux, Daneshwar Puchooa, Nadeem Nazurally, Abdulwahed Fahad Alrefaei, Ying Zhang

**Affiliations:** 1Department of Health Sciences, Faculty of Medicine and Health Sciences, University of Mauritius, Réduit 80837, Mauritius; 2Department of Zoology, College of Science, King Saud University, P.O. Box 2455, Riyadh 11451, Saudi Arabia; 3Department of Agricultural and Food Science, Faculty of Agriculture, University of Mauritius, Réduit 80837, Mauritius; 4School of Ecology and Natural Conservation, Beijing Forestry University, 35 East Qinghua Road, Haidian District, Beijing 100083, China

**Keywords:** microorganisms, marine, polysaccharides, diversity, biotechnological applications

## Abstract

As the largest habitat on Earth, the marine environment harbors various microorganisms of biotechnological potential. Indeed, microbial compounds, especially polysaccharides from marine species, have been attracting much attention for their applications within the medical, pharmaceutical, food, and other industries, with such interest largely stemming from the extensive structural and functional diversity displayed by these natural polymers. At the same time, the extreme conditions within the aquatic ecosystem (e.g., temperature, pH, salinity) may not only induce microorganisms to develop a unique metabolism but may also increase the likelihood of isolating novel polysaccharides with previously unreported characteristics. However, despite their potential, only a few microbial polysaccharides have actually reached the market, with even fewer being of marine origin. Through a synthesis of relevant literature, this review seeks to provide an overview of marine microbial polysaccharides, including their unique characteristics. In particular, their suitability for specific biotechnological applications and recent progress made will be highlighted before discussing the challenges that currently limit their study as well as their potential for wider applications. It is expected that this review will help to guide future research in the field of microbial polysaccharides, especially those of marine origin.

## 1. Introduction

Over the past few years, there has been increasing interest in the development of natural polymers, also referred to as biopolymers, for industrial applications [1], and in particular, polysaccharides have been gaining much attention in the biomedical, cosmetic, food, and pharmaceutical fields. Although polysaccharides can be produced by different types of organisms (e.g., bacteria, fungi, algae, crustaceans, and plants), those from bacteria and fungi have been highly popular as they replicate rapidly, are easier to manipulate, and are abundant producers of those polymers, with the latter also more easily separated than those from non-microbial counterparts [2,3]. Furthermore, in addition to their biological activities, they also display low toxicity, biocompatibility, biodegradability, and other physical and chemical characteristics [4,5,6,7]. Thus, without undermining the value of non-microbial sources of polysaccharides, this review mainly focuses on those obtained from bacteria and fungi in order to highlight their potential for future studies, especially in view of developing polymers for practical applications. 

An overview of these microbial compounds suggests that they are usually macromolecules of high molecular weight and are made up of at least 10 monosaccharide units held together by glycosidic bonds [8]. They can also be either linear or branched in structure, while in terms of content, they may be classified as homopolysaccharides or heteropolysaccharides if they, respectively, contain one type or different types of monosaccharides [7,8,9]. In the latter case, while D-glucose is often the most common constituent, other sugars (e.g., D-xylose, D-mannose, D-galactose, L-galactose, L-arabinose, and D-fructose) may also be present alongside derivatives such as simple sugar acids (glucuronic and iduronic acids) or amino sugars (D-glucosamine and D-galactosamine) [10]. Finally, in addition to the above sugars, the presence of non-organic moieties, including sulfates, phosphates, pyruvates, and acetates, is also frequently noted [4,5]. Altogether, different combinations of these variable features contribute to the extensive compositional and structural diversity displayed by microbial polysaccharides.

Given that the structure of polysaccharides ultimately determines their functions, such diversity is actually of functional significance to microorganisms. Indeed, through variations in the monosaccharide composition, non-sugar side chains, glycosidic linkages, and other characteristics, bacteria and fungi are able to synthesize a wide range of polysaccharides that are involved in different biological processes [8]. For instance, some polysaccharides, such as glycogen, occur intracellularly, where they act as storage molecules to provide energy under starvation conditions [4,8,11]. Others are part of bacterial and fungal cell walls or capsules, and as structural polymers, they not only are responsible for maintaining cellular integrity but also assist in other functions such as conferring protection against environmental stresses, regulating membrane permeability, or even mediating interactions with the surroundings, which, in the case of pathogenic microorganisms, may include the onset of immunological responses [12,13,14]. Finally, there are polysaccharides that are secreted outside microbial cells and are thus often termed as exopolysaccharides or extracellular polysaccharides (EPSs). Being commonly synthesized by bacteria and fungi, EPSs are arguably one of the most studied ones, as reflected in the number of publications available on the subject. This can probably be attributed to the fact that, unlike intracellular or cell wall polysaccharides, EPSs are produced in relatively larger amounts within a short time while being more easily isolated and purified. As such, they are better suited for practical applications and will be the main subject of focus in this review [15,16].

## 2. The Case of Marine Polysaccharides

With around 70% of the Earth’s surface covered with water, the marine environment is undoubtedly an attractive source of microbial polysaccharides [17]. Indeed, aquatic habitats are known to harbor a large diversity of microorganisms, and, in a similar way to terrestrial microbes, it is likely that this also translates into significant diversity in terms of the polysaccharides that they can produce [18], as evidenced by previous reports (e.g., [5,19]). However, increasing interest in marine polysaccharides over the past few years can arguably be attributed to the potential of isolating polysaccharide-producing organisms that can be sources of novel polymers. Indeed, it is often reported that fewer than 1% of marine microorganisms are currently known or cultured, and as a result, the microbial populations in different marine ecosystems remain relatively under-explored [20,21,22]. Therefore, there is an increased likelihood of identifying novel microorganisms, which, in turn, may lead to the isolation of novel polysaccharides with unique properties, especially since some taxa can produce only specific polymers [4]. For instance, in India, Srivastava et al. [23] isolated a novel heteropolysaccharide of around 286 kDa, made up of glucose and galacturonic acid, from the marine bacterium *Brevibacillus borstelensis*, while in a different study, *Rhodotorula mucilaginosa*, a marine-derived red yeast, was found to yield a new 1200 KDa exopolysaccharide that consisted of fucose, galactose, mannose, and glucose [24]. Similarly, there are also reports on the isolation of novel exopolysaccharide-producing species or other new marine-derived polymers [25,26,27,28,29]. Table 1 provides some examples of novel polysaccharides and/or novel polysaccharides-producing marine bacteria and fungi that have been reported over the past decade. While such examples may not be exhaustive, they clearly highlight the potential of exploring aquatic species in the search of new polysaccharides.

Furthermore, as already pointed out, polysaccharides are often produced to protect bacteria and fungi from their surroundings. Hence, in response to the unique conditions prevailing in marine environments, it can be expected that aquatic species may develop specific metabolic and physiological capabilities for better adaptation, thereby resulting in the production of compounds, including polysaccharides, which may be absent from terrestrial microbes [48]. This was highlighted in the study by Abdel-Wahab et al. [37], in which a marine strain of the fairly common *Bacillus subtilis* species yielded a novel β-glycosidic sulfated heteropolysaccharide. This polymer, consisting of glucose, rhamnose, and arabinose, could also exhibit a wide range of biological activities (anti-oxidant, anti-inflammatory, cytotoxicity, and anti-Alzheimer activities). Such protective functions have been of particular value in studies involving microorganisms from extreme marine habitats. Indeed, some species are able to survive in specific areas characterized by very high or low temperatures (thermophiles and psychrophiles), high or low pH (acidophiles and alkalophiles), high pressures (piezophiles), or even high ionic strengths (halophiles) [49]. In these cases, the extremophiles adopt specific survival strategies, which include but are not limited to the production of polysaccharides with unique properties [50,51], with examples of such polysaccharides that have been isolated from extremophilic microorganisms during the last decade, provided in Table 2. Thus, it can be expected that the study of microbial species from extreme environments could yield polysaccharides with new or improved properties.

An overview of the above tables suggests some common features in the study of microbial polysaccharides. Firstly, the high diversity of novel bacterial and fungal polysaccharides is quite obvious, especially in terms of the molecular weight, composition, and biological activities, all of which further lend support to the potential of exploring such microbial compounds. However, although the above list is not exhaustive, there seems to be a greater focus on polysaccharides from marine bacteria as compared to fungal sources, probably due to the former’s higher diversity as free-living organisms, as well as their ease of isolation in terms of growth requirements. This is also particularly obvious as far as polymers from extremophiles are concerned. Nevertheless, fungal sources remain a major source of polysaccharides, and research on such species is still likely to yield polysaccharides of interest. In addition, in terms of the level of characterization, determining the molecular weight, functional groups, and monosaccharide composition tend to be standard practices, but interestingly, a survey of the literature suggests that structural characterizations, even partial ones, seem to also be gaining in importance. However, studies that fully establish structure–function relationships for polysaccharides in view of explaining their biological activities are still far from common, probably because a highly technical and experimental analysis is required to determine the structures of such high molecular weight polymers [37].

## 3. Current Research on Marine Microbial Polysaccharides

Over the years, the potential of microbial polysaccharides for biotechnological applications became increasingly recognized, as reflected by the number of studies on the subject, with Osemwegie et al. [2], as reported by Nadzir et al. [57], identifying thousands of publications focused on microbial polysaccharides between 1976 and 2018 alone. Although many of these did not specifically involve marine species, a survey of the literature suggests a similar trend as far as marine microbial polysaccharides are concerned. The following sections provide an overview of the current trends in research on marine polysaccharides from bacterial and fungal species.

### 3.1. Biomedical Applications

One of the most promising properties displayed by marine microbial polysaccharides is their biological activities. Indeed, it is not uncommon for studies to investigate such characteristics in view of presenting these polymers as attractive candidates for biomedical applications, with some of the most studied biological activities being anticancer, antimicrobial, anti-oxidant, and immunomodulation [18].

#### 3.1.1. Anticancer Activity

Cancer, characterised by an uncontrolled proliferation of cells, is currently one of the major diseases affecting human health, as well as the second cause of death in the world, with an estimated 18.1 million people diagnosed with the condition in 2018 [58,59]. So far, surgery, chemotherapy, and radiotherapy remain the main forms of treatment for different types of cancers, but these approaches are not without side effects, which include a number of health complications as well as toxicity and/or injury to non-targeted organs and cells [58,60,61]. Consequently, the search for alternative forms of treatment has prompted interest in natural compounds, with results of studies often highlighting the potential of marine microbial polysaccharides for such applications.

Indeed, the anticancer activities of polysaccharides from marine bacteria and fungi are already well established, with cytotoxic effects often reported against lung [32], liver [29,31,62,63], breast [59,62,64,65,66], cervical [62], and colorectal cancers [59]. In these cases, the apoptosis of cancer cells seems to be a common mechanism, although programmed cell death can be mediated through different pathways. These different mechanisms of action are particularly obvious from studies in which different polysaccharides were tested against different cell lines. For example, Tukenmez et al. [67] noted changes in the gene and protein expression of Bax, Bcl-2, Caspase 3, Caspase 9, and Survivin when EPSs of *L. delbrueckii* ssp. *Bulgaricus* were tested against colon cancer cells at a concentration of 400 µg/mL for 24 h or 48 h. The EPS consisted of glucose, mannose, fructose, sucrose, maltose, and N-acetylglucosamine, with the observed effects attributed to the glucose and mannose content. In contrast, within a concentration range of 5–80 μg/mL, marine polysaccharides from *Bacillus velezensis* activated caspase-3 while increasing levels of cytochrome C to induce apoptosis in breast cancer cells [65]. Such differences in the mechanism of action could arguably be attributed to differences in the composition and/or structure of the two polysaccharides, as suggested by Tukenmez et al. [67], although the influence of other factors is not excluded. In addition, within similar cell lines, anticancer effects may also occur through different mechanisms. For instance, Cao et al. [32] reported the isolation of EPS11, a 22.3 kDa polysaccharide fraction made up of glucose, mannose, xylose, glucosamine, and galacturonic acid, from a marine *Bacillus* species. At varying concentrations up to 90 nM, the authors showed that this polymer could not only affect cell proliferation and adhesion of lung (A549) and liver (Huh7.5) cells, but also induce apoptosis by preventing the expression of βIII-tubulin, as well as reducing the phosphorylation of protein kinase B (PKB or AKT). However, the same polysaccharide also downregulated proteins related to the extracellular matrix–receptor interaction signaling pathway and targeted collagen I through the β1-integrin-mediated signaling pathway to prevent cell adhesion, migration, and invasion [31]. This feature of polysaccharides is, in fact, of practical significance, as the ability to induce anticancer effects through different mechanisms may reduce the likelihood that a particular cell line develops resistance to therapy, as is currently the case for a number of chemotherapeutic drugs [68]. 

#### 3.1.2. Antimicrobial Activity

Another biological activity of microbial polysaccharides that is commonly investigated is their ability to inhibit the growth and/or proliferation of pathogenic microorganisms. This property is particularly relevant nowadays due to the emergence of drug-resistant pathogens that constantly threaten public health, thereby prompting the need to develop new and more potent antibiotics [69,70]. In this context, Aullybux et al. [71] isolated two sulfated EPSs from a marine *Alcaligenes* and *Halomonas* sp., which could, in addition to different pathogens, also inhibit the growth of methicillin-resistant *Staphylococcus aureus* (MRSA) at concentrations between 0.25 and 2 mg/mL. Although not specifically attributed to any particular features of the polymers, the authors argued that the antibacterial activities could be linked to the presence of certain functional groups that are known to act as metal chelators. Similarly, polysaccharides (1 mg/mL) from a haloalkalitolerant *Alkalibacillus* sp., recovered from a salt lake, displayed antibacterial effects against *Candida albicans,* as well as a number of Gram-positive and Gram-negative bacteria [72].

Although these studies did not determine the underlying mechanism of the antimicrobial properties, this can be inferred based on existing reports from other non-marine polysaccharides. For example, electrostatic interactions between oppositely charged polysaccharides and pathogens’ cell walls, as well as the latter’s subsequent hydrolysis to leak cell content, have been suggested as one of the mechanisms responsible for the observed antimicrobial activities [71]. Similarly, Zhou et al. [73] proposed that interactions between polysaccharides and biofilm-related signal molecules or cell-surface receptors of pathogens could disrupt cell communication and biofilm formation, while Rajoka et al. [74] suggested that metal chelation, as well as nutrient suppression through the formation of an external barrier, could represent additional ways through which antimicrobial properties are exerted.

However, while the antimicrobial properties of bacterial and fungal polysaccharides are well known, those derived from marine species are yet to be widely studied, as is the case for their terrestrial counterparts. As will be discussed in subsequent sections, this could likely be due to the different challenges encountered in the study of marine microbial polysaccharides. Nevertheless, this undoubtedly represents a research gap that, if addressed, could potentially yield new classes of antibiotics to assist the fight against resistant pathogens [75].

#### 3.1.3. Anti-Oxidant Activity

Bacterial or fungal polysaccharides, especially those that display biological activities, have been considered not only for their cytotoxicity or antibacterial effects but also for their anti-oxidant potential [76,77]. Nowadays, it is not uncommon to come across studies that highlight the anti-oxidant potential of compounds, probably because it is intrinsically linked to human health. 

Tissues in the human body require oxygen for energy production, but as the oxygen is consumed, those tissues generate free radicals as by-products [78]. Free radicals are, basically, reactive and unstable molecules containing unpaired electrons, and they can be grouped into either reactive nitrogen species (RNSs) or reactive oxygen species (ROSs) [78,79]. These compounds are normally maintained at a suitable concentration by balancing the body’s production of free radicals through a defence system involving anti-oxidant enzymes [80]. However, in addition to cellular production, external sources such as radiation, chemicals, pollutants, cigarettes, alcohol, some drugs, or heavy metals, just to name a few, can also contribute substantially to the levels of ROS/RNS [80]. These can undoubtedly cause an imbalance, especially if the body is no longer able to counteract the additional production of free radicals, hence resulting in a condition of oxidative/nitrosative stress. Under this condition, these excess radicals, especially ROS, interact with various biological macromolecules such as proteins, DNA, RNA, and lipids, causing their structural and functional alterations [80,81]. Given that these macromolecules have important physiological functions, it is, therefore, not surprising that oxidative/nitrosative stress has been established as a major cause of human diseases, which include cardiovascular, organ disorders (pancreas, lungs, eyes, kidneys, and joints), neurodegenerative diseases, and even cancer [78,80].

In light of the above, it can be understood why anti-oxidant compounds are highly regarded as being beneficial for health. Indeed, anti-oxidants are molecules that, at low concentrations, inhibit or cause a significant delay in the oxidation of compounds [82], and as such, they help to hinder the negative effects of oxidative stress. Interestingly, reports on the anti-oxidant activities of polysaccharides from marine bacteria and fungi are not lacking, with some even involving novel species (e.g., *Enterobacter cloacae* MBB8) or novel polymers (e.g., EPSR4 from *Bacillus subtilis*) [25,37,83,84]. However, while this commonly studied activity of marine microbial EPS is now fairly established, the next step would undoubtedly be developing practical applications for such polymers. In this context, it is worth noting that providing anti-oxidants as supplements has been suggested as a means of mitigating the negative effects associated with oxidative stress [85]. Hence, the successful isolation of anti-oxidant EPS from two probiotic marine bacteria (*Rhodotorula* sp. and *Pediococcus pentosaceus*), as reported by Wang et al. [86] and Ayyash et al. [87], further highlights the potential of studying marine microorganisms for this purpose.

#### 3.1.4. Drug Delivery

Besides their biological activities, microbial polysaccharides are also suitable to enhance the activities of other compounds, and for this purpose, they are often applied in the development of drug-delivery systems. Such nano-based or targeted delivery of therapeutic agents ensures that the latter are delivered at the required site and in a controlled manner, thereby overcoming the limitations (e.g., drug bioavailability, unwanted side effects, and non-specificity of drugs) that are encountered with current methods of drug delivery [88,89]. For this purpose, the selection of a suitable carrier molecule is a key parameter that needs to be considered, as its properties eventually influence the mechanism of drug release [90]. However, more importantly, the selected carrier would need to be biodegradable, biocompatible, and safe in order to be considered for such applications, and interestingly, microbial polysaccharides display such characteristics [90,91].

Dextran, composed of a linear chain of D-glucose linked by α-(1→6) bonds and commonly synthesized by lactic acid bacteria, is a widely used EPS for developing targeted delivery systems [92]. As a result of its non-toxicity, non-immunogenicity, and biocompatibility, dextran represents a suitable polymer for encapsulating or adsorbing therapeutic agents and ensuring their delivery while effectively providing protection against the immune system, as well as digestive enzymes [93]. The efficacy of such systems was demonstrated by Wang et al. and Fang et al., who developed dextran-based nanocarriers for the delivery of doxorubicin, with the results confirming the improved anticancer effects alongside reduced toxicity to the drug [94,95]. Similarly, chitosan (a derivative of chitin) and levan (a fructose homopolymer) also display properties such as biocompatibility, biodegradability, and low toxicity [96,97]. These features were further confirmed by studies whereby cisplatin, 5-fluorouracil, and resveratrol were successfully loaded onto those polysaccharides for delivery to cancer cells while being safe for healthy ones [98,99,100]. 

While there are reports on the production of the above polysaccharides from marine microorganisms (e.g., *Penicillum spinulosum* and *Halomonas* sp.) [101,102], it is surprising to note that such applications are yet to be established for those obtained from marine bacteria or fungi. This was reflected in an overview of two recent reviews on the application of marine microbial EPS as drug carriers, whereby the focus was largely, if not completely, on polysaccharides from non-marine sources [90,91]. However, the absence of a significant number of studies on the subject does not suggest that the potential of marine microbial polysaccharides has been overlooked. For instance, in an attempt to develop microgels as protein carriers, Zykwinska et al. successfully assembled EPSs from *Vibrio diabolicus*, a deep-sea hydrothermal bacterium, for the encapsulation of bovine serum albumin [103]. As a possible extension to that study, the authors subsequently isolated EPS from *Alteromonas infernus* to yield microcarriers that could encapsulate Transforming Growth Factor-β1 (TGF-β1) for applications in cartilage engineering [104]. Similarly, K1^T^-9, a 207 kDa heteropolysaccharide isolated from the novel marine bacterium *Neorhizobium urealyticum*, was successfully applied as an emulsifier for the encapsulation of astaxanthin [39]. Therefore, given the diversity and possibly unique properties of marine microbial polysaccharides, addressing the current research gap could potentially lay the foundations for the development of new or improved drug-delivery systems based on these polymers.

### 3.2. Bioremediation

Over the past few decades, industrialization and other human activities such as the improper disposal of wastes and the excessive use of pesticides and fertilizers have been a major cause of environmental pollution [105]. In particular, water contamination via heavy metals such as chromium (Cr), cadmium (Cd), lead (Pb), arsenic (As), mercury (Hg), and silver (Ag), just to name a few, has been of concern not only due to their non-biodegradability and toxicity at certain concentrations but also due to their potential accumulation along the food chain [106]. Therefore, the removal of heavy metal contaminants has been devised as an effective strategy for treating polluted areas, and in this context, electroplating, ion exchange, precipitation, and membrane processes represent some of the most commonly used approaches for this purpose [106,107]. However, these methods are not without drawbacks, the most important of which include the high cost involved, its low efficiency, and the production of toxic by-products [106]. As a result, attention has shifted to better alternatives, with microbial-based treatments proving to be a suitable candidate for such applications.

Indeed, microorganisms, especially those from heavy-metal-polluted areas, have evolved to develop tolerance to such pollutants, and therefore, they can be ideal candidates for bioremediation processes [108,109]. While the microbial-based removal of toxic heavy metals from the environment can be mediated through different pathways [109], the current review will focus on the potential of their polysaccharides for this purpose. Microbial polysaccharides contain a number of functional groups such as carboxyl, hydroxyl, phosphate, amine, and uronic acids, with marine-derived ones being particularly rich in the latter [5,110]. These groups confer an overall negative charge to the polymers, thereby allowing them to bind to the positively charged heavy metals through the process of adsorption and subsequently leading to their removal [5]. For instance, a *Bacillus cereus* strain, isolated from a contaminated estuarine sediment, showed potential for water detoxification at concentrations of 25 to 150 mg/L due to its EPSs’ affinity for Pb, Cu, and Cd [106]. In this case, higher adsorption capacity, largely attributed to the functional groups present, occurred at the lower doses. Similarly, Concórdio-Reis et al. [111] reported the isolation of the EPS FucoPol from an *Enterobacter* species. This polymer, which showed specificity towards Pb, had an overall metal removal efficiency of 91.6–93.9% and this was achieved at a concentration of 5 g/L through its carboxyl and hydroxyl groups, under acidic conditions and within a temperature range of 5–40 °C. A different study further highlighted the potential of marine environments as a source of novel polysaccharides for bioremediation processes. Indeed, Zhang et al. [38] reported the ability of a novel polymer from an *Alteromonas* species to adsorb Cu, Ni, and Cr at a concentration of 1 g/L, thus indicating its suitability for the removal of heavy metals. However, despite its novel structure, the observed effects were still attributed to the functional groups present, especially O-H, C=O, and C-O-C, as is often the case for other microbial polysaccharides.

Closely related to the above are potential applications of marine microbial polysaccharides as bioflocculants. Flocculation refers to the aggregation of small suspended particles into larger flocs to aid their removal, and while it is not limited to heavy metals, it is nevertheless also applied to water treatment [112]. In this context, Chen et al. characterized a novel bioflocculant from the marine *Alteromonas* species [113]. Largely composed of polysaccharides, this polymer (20–220 mg/L) could effectively remove dyes such as Methylene Blue, Direct Black, and Congo Red at efficiencies between 72.3% and 98.5%, thus proving to be effective for the treatment of dyed wastewater. In a different study, a marine *Bacillus* species achieved 85% bioflocculant activity, with the optimum conditions for such activities also determined [112]. Overall, based on the above studies, it would not be unlikely that the huge diversity of marine microbial polysaccharides, as well as their specificity to pollutants, could drive the search for new polymers. At the same time, the fact that the presence of a polysaccharide backbone can enhance the thermal stability of bioflocculants could spark additional interest in the isolation of such polymers.

Although the above examples are not exhaustive, a key factor that explains the wide interest in polysaccharides is the remarkable structural diversity displayed by these polymers, as this translates into a wide range of properties, as well as potential applications. Thus, any prospective uses of polysaccharides are often dependent on structural characteristics such as the monosaccharide composition, the type of functional groups present, or even the conformation, including the degree of branching and the type of linkage [114]. For instance, the presence of neutral monosaccharides (e.g., glucose, mannose, fucose, arabinose, D-galactose, and glucuronic acid) has been reported as being more likely to induce anti-oxidant activities [76], while in their study on polysaccharides from *Lactobacillus reuteri*, Chen et al. [115] noted that the amount of galactose was related to the anti-inflammatory activity of the polymer, with higher amounts of the monosaccharide enhancing the biological activity. Similarly, in terms of conformations, the types of glycosidic linkages may affect the solubility and flexibility of polysaccharide chains, thus making them more or less suited to certain specific applications [67,76]. In other cases, polymers with mostly β-1,3-linkages were also reported as displaying greater antitumor activities in contrast to those containing mostly β-1,6- linkages [67]. Finally, as far as functional groups are concerned, biological activities are often observed when specific groups are present. For example, phosphate groups can contribute to the immunomodulatory effects of polysaccharides by improving their affinity to immune cells. In addition, compared with neutral polysaccharides, phosphorylated ones are also more likely to inhibit the growth of certain cancer cells [116,117]. Similarly, sulfated or acetylated polysaccharides can display better biological activities than non-sulfated ones, especially antibacterial, anti-oxidant, and antitumor effects [116,118,119]. As discussed in the subsequent section, the influence of the functional groups on the biological properties of polysaccharides is actually of great significance in generating specific polymers of practical value.

## 4. Challenges for the Commercialization of Marine Microbial Polysaccharides

Despite the wide interest in microbial polysaccharides, as well as their great potential for a number of applications, only a few of these polymers have actually been commercialized so far, with even fewer being from marine microorganisms (e.g., HE800 EPS, bearing the trademark Hyalurift^®^ [120], and HYD657, commercially available as Abyssine^®^ [121]). However, a survey of deposited patents related to marine microbial polysaccharides does suggest that the potential of these polymers for a wide range of applications is recognized (Table 3). Hence, while learning the barriers to commercialization may not necessarily result in more products, addressing them may potentially increase the likelihood that the results of research are eventually translated into practical applications. 

One of the main factors that limit the production of polysaccharides is the high cost of production, even though microorganisms are relatively easier to grow and manipulate, are better producers, and are, overall, cheaper sources of biopolymers in comparison with non-microbial sources [5,122]. In this case, improving the yield of polysaccharides is a commonly applied strategy to make the process more cost-effective, and this can often be achieved through process optimization, whereby the growth conditions that maximize yields are identified and applied. For instance, Hereher et al. managed to increase EPS production from a *Micrococcus roseus* strain by over four times by modifying the amount of sucrose and ammonium sulfate, as well as the incubation temperature and pH [123]. Similarly, a three-fold increase in EPS yield was reported for a *Halomonas xianhensis* strain by optimizing culture conditions [124]. In addition, favoring the use of cheaper substrates has also been proposed as a means of bringing down the overall cost of production [125,126]. 

However, it should be noted that it is not uncommon for some microbial species to still have relatively low yields despite process optimization, and, in this case, the genetic modification of the microorganisms can provide a powerful alternative to improve yields, especially since they are more easily amenable to genetic changes compared to higher organisms [127]. Such modifications can range from increasing the availability of EPS precursors, overexpressing genes involved in EPS assembly, or even knocking out those that compete against EPS production. For instance, the marine yeast *Aureobasidium melanogenum* was successfully modified with the *INU1* gene to improve pullulan production by more than five times compared with the wild strain [128]. At the same time, in addition to better yields, this approach may also help in generating tailor-made polysaccharides with better or novel physicochemical and biological properties [129]. Interestingly, as pointed out by Wang et al. [130], these benefits can be particularly suited to extremophilic microorganisms, which, despite their potential to synthesize EPS with distinct features and properties, are relatively poor producers of such polymers.

Another commonly applied strategy that has been proposed as a means of improving the value of isolated microbial polysaccharides is structural modification. As noted before, the properties of microbial polysaccharides are tightly linked to their structures, and therefore, this approach is particularly suited to cases where the isolation of closely related microbial species yields polysaccharides of nearly similar structures or those with no biological activities, both of which may not bring significant additional value to existing research [9,131]. In other cases, although polysaccharides may display the potential for specific applications, the absence of suitable physico-chemical properties (e.g., poor solubility) may hinder subsequent interest in their exploitation [50]. Thus, by altering the structures of such polymers through physical, biological, and chemical means, their existing properties can be enhanced and tailored for individual applications [50,131]. For instance, through a sulfation modification, Chopin et al. [132] improved the ability of GY785, an EPS from the bacterial strain *Alteromonas infernus* isolated from deep-sea hydrothermal vents, to drive the chondrogenic differentiation of mesenchymal stem cells, hence enhancing its potential for cartilage repair. Furthermore, there are also reports on phosphorylation modifications, which not only enhance existing biological activities of polysaccharides but also help to induce such activities in polymers in which they are naturally absent [131,133]. Overall, while reports of such modifications of polysaccharides from marine bacteria and fungi are not extensive, it is likely that this approach can be a powerful tool that could increase the likelihood of developing commercially viable marine microbial polymers. However, for most reported microbial polysaccharides, the relationship between structure and biological functions is not precisely established, hence making it more difficult for commercialization [57]. Therefore, the above examples clearly highlight the need to characterize the structures of microbial polysaccharides as part of studies, especially since such information would provide an understanding of potential modifications that could be undertaken in view of obtaining desirable properties.

According to Li et al. [58], the low application of polysaccharides can also be linked to the lack of in vivo research. Although studies involving animal models are not completely absent, integrating more in vivo research is more likely to provide a better insight into the actual potential of polysaccharides, especially as far as their biomedical applications are concerned. Finally, as noted before, most marine microorganisms remain unculturable, and this clearly limits studies on marine polysaccharides, as these require culturable samples for extracting and characterizing the polymers [134]. Therefore, devising new culture methods to improve the isolation of previously uncultured microorganisms can pave the way to the production of novel polysaccharides.

## 5. Conclusions and Future Perspectives

This paper provides an overview of marine microbial polysaccharides, their peculiarities, and current trends in their study. Research gaps, including challenges to their practical applications, were also briefly addressed. With much of the marine environment still relatively unexplored, there is no doubt that research on marine microbial polysaccharides remains promising. In particular, focusing on extreme environments may yield polymers which are not only unique in terms of their properties but also of potential high commercial value. In fact, as talks on climate change and the need to reduce petroleum-derived compounds step up, switching to sustainable alternatives is likely to take on even greater importance, thereby further fuelling interest in marine microbial polysaccharides. 

## Figures and Tables

**Table 1 marinedrugs-21-00420-t001:** Examples of novel polysaccharides- and/or novel exopolysaccharides-producing marine bacteria and fungi, reported over the past decade, that highlight the large diversity of these polymers in terms of size, composition, and biological activities. (* tr—trace amounts; N/A—not available)—Note: when not available, figures of polysaccharide structures were inferred based on information provided in the relevant studies and should not be taken as representing the actual or most stable conformations of the polymers.

Organism Name	Composition	Molecular Weight (kDa)	Possible Structure(s)	Biological Activity (If Any)	Level of Characterization	Reference
*Vibrio alginolyticus*	Mannose, glucosamine, gluconic acid, galactosamine and arabinose (5:9:3.4:0.5:0.8)	14.8	N/A	Antitumor activity	Molecular weight, monosaccharide composition, functional groups, surface morphology, and element composition	[30]
*Bacillus* sp.	Mannose, glucosamine, galacturonic acid, glucose and xylose (1:2.58:0.68:0.13:3.09:1.41)	22.3	N/A	Anticancer through different mechanisms	Molecular weight and monosaccharide composition	[31,32,33]
*Microbacterium aurantiacum*	Glucose, mannose, fucose and glucuronic acid	7000	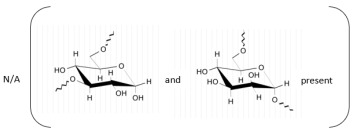	Anti-oxidant activity and viscosifying property	Molecular weight, total sugar, total protein and uronic acid content, functional groups, element analysis, monosaccharide composition, and partial structure	[34]
*Rhodobacter johrii*	Glucose, galactose, rhamnose and glucuronic acid (3:1.5:0.25:0.25)	2000	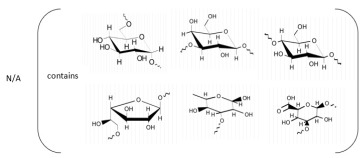	Bioemulsification property	Molecular weight, total sugar, total protein and uronic acid content, functional groups, element analysis, monosaccharide composition, and partial structure	[35]
*Pseudoalteromonas* sp.	Mannose, glucose, galactose, rhamnose, xylose, N-acetylgalactosamine and N-acetylglucosamine	>2000	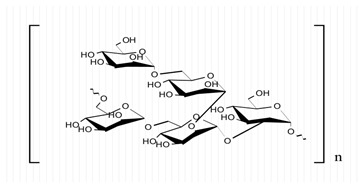	-	Molecular weight, total sugar content, monosaccharide composition, linkage analysis, and structure	[36]
*Bacillus subtilis*	Glucose, rhamnose and arabinose	14.8	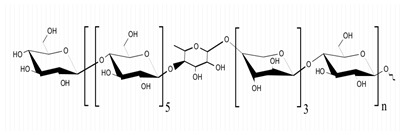	Anti-oxidant, anti-inflammatory, cytotoxicity, and anti-Alzheimer activities	Molecular weight, functional groups, uronic acid and sulfate content, monosaccharide composition, surface morphology, and crystal structure	[37]
*Alteromonas* sp.	Rhamnose, mannose and galacturonic acid	167	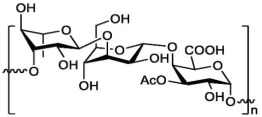	Biosorption of heavy metals	Molecular weight, total sugar content, functional groups, linkage analysis and structure	[38]
*Neorhizobium urealyticum* K1^T^ sp. nov.	Glucose and galacturonic acid	207	N/A	Emulsification	Molecular weight, total sugar and total protein content, functional groups, element analysis, monosaccharide composition, and partial structure	[39]
*B. licheniformis*	Fructose, fucose, glucose, galactosamine and mannose (1.0:0.75:0.28:tr:tr) *	1000	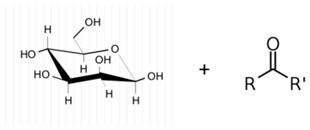	Cytotoxic, antiviral and immunomodulatory properties	Molecular weight, monosaccharide composition, polar lipids and fatty acids content, and structure	[40,41]
*Pantoea* sp.	Glucose, galactose, N-acetyl galactosamine and glucosamine (1.9:1:0.4:0.02)	175	N/A	Cutaneous wound healing	Molecular weight, monosaccharide composition, total sugar and total protein contents, and functional groups	[42]
*Bacillus* sp.	Mannose, glucosamine, glucose, and galactose (1.00:0.02:0.07:0.02)	89	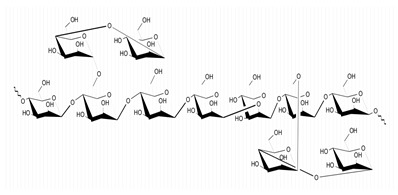	Immunomodulation	Molecular weight, monosaccharide composition, linkage analysis, functional groups, and structure	[43]
*Natronotalea sambharensis* sp. nov.	Mannose, glucose and glucuronic acid	4.6 × 10^6^	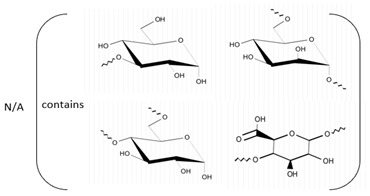	Anti-oxidant	Molecular weight, total sugar, total protein, nucleic acid and uronic acid content, element analysis, functional groups, surface morphology, monosaccharide composition, and partial structure	[44]
*Aspergillus ochraceus*	Mannose and galactose (2.16:1.00)	29	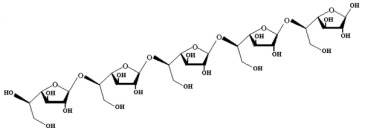	-	Molecular weight, total sugar, protein and uronic acid content, monosaccharide composition, linkage analysis, sugar configuration, and structure	[45]
*Aspergillus versicolor*	Glucose	5.1	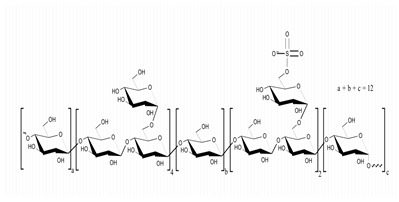	Immunomodulation	Molecular weight, functional groups, monosaccharide composition, and linkage	[46]
*Penicillium janthinellum*	Mannose and galactose	10.24	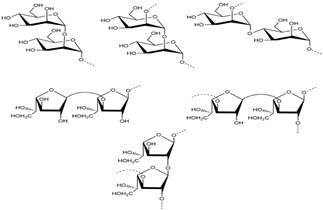	Anti-diabetic activity	Molecular weight, total sugar and total protein content, functional groups, monosaccharide composition, linkage analysis, sugar configurations, and structure	[47]

**Table 2 marinedrugs-21-00420-t002:** Examples of polysaccharides isolated from extremophilic microorganisms.

Species Name	Type of Environment	EPS Name	EPS Composition	Distinguishing Features	Reference
*Polaribacter* sp.	Polar region (Arctic)	EPS	N-acetyl glucosamine, mannose, glucuronic acid, moderate amounts of galactose and fucose, and minor amounts of glucose and rhamnose	Tolerance to high salinity and a wide pH range	[52]
*Alteromonas infernus*	Deep-sea hydrothermal vent	GY785	Glucose, galactose, galacturonic acid and glucuronic acid	-	[53]
*Vibrio diabolicus*	Deep-sea hydrothermal vent	HE800	N-acetyl glucosamine, N-acetyl galactosamine and glucuronic acid	-	[53]
*Pseudomonas* sp.	Polar region (Antarctica)	EPS	Glucose, galactose, fucose, and uronic acid	Cryoprotection and emulsification	[54]
*Zunongwangia profunda*	Deep-sea (1245 m)	EPS	-	High moisture retention and anti-oxidant potential	[55]
*Halomonas nitroreducens*	Hydrothermal vent	EPS	Three different EPSs made up of glucose, mannose, galactose, and small quantities of rhamnose, arabinose, and galacturonic acid in variable amounts	Pseudoplastic nature with high emulsifying, anti-oxidant, and heavy metal-binding activities	[56]

**Table 3 marinedrugs-21-00420-t003:** Examples of patents that have been deposited for polysaccharides derived from marine bacteria or fungi during the last decade.

Patent Number	Species	Source	Characteristic of Polysaccharide	Patented Application
AU2016330332B2	*Alteromonas* sp.	Deep-sea hydrothermal environment	15 kDa over-sulfated exopolysaccharide (GYS15)	Anti-metastatic and/or related uses for various cancers
CN116120477A	-	Antarctic sea	4350–4360 kDa extracellular polysaccharide with low temperature resistance and moisturizing functions	Preparation method and application
ES2585398B1	*Pseudomonas* sp.	Marine sediment	2000 kDa exopolysaccharide with cryoprotective, emulsifying, thickening, stabilizing, or texturing properties	Cosmetic application
CN105087450A	*Alteromonas marina*	-	167 kDa exopolysaccharide	Culture of organism and preparation of polysaccharide
US10993434B2	*Pseudoalteromonas* sp.	Polar region	100–430 kDa exopolysaccharide	Cryoprotection of cells
CN107523515A	*Pseudoalteromonas* sp.	-	-	Absorption of heavy metals from drinking water
LU501700B1	*Aerococcus urinaeequi*	-	-	Growth of microorganism and polysaccharide production
CN109457001A	*-*	-	Polysaccharide with mannose, glucosamine, ribose, rhamnose, glucuronic acid, galacturonic acid, glucose, galactolipin, xylose, and arabinose	Preparation and application as decoloring agent

## Data Availability

Not applicable.

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
