# Peer review of "Marine Microbial Polysaccharides: An Untapped Resource for Biotechnological Applications"

_marinedrugs, 2023, doi:10.3390/md21070420_

Round 1

Reviewer 1 Report

The article of R. Jeewon and co-workers untitled “Marine microbial polysaccharides: An untapped resource for biotechnological application” is a short review dealing with the potential applications of marine bacterial polysaccharides and more especially those secreted by the microorganisms: the exopolysaccharides (EPS).

General comments:

The article which aims at giving an overview of the state-of the art and the potential of marine microbial polysaccharides dramatically lacks of illustrations, quantification and support of the statements. For example, in section 2 “The case of marine polysaccharides”, the authors didn’t give any element on the state of art concerning the number of new polysaccharides isolated which composition and structure have been determined the last decade (or in general). Therefore, it is difficult to appreciate what effort is involved by the scientific community on this topic.

In this vein, the section 3 “Current biomedical research on marine microbial polysaccharides” lists applications of the EPS illustrated with examples but don’t give any information on the level of characterization of the polysaccharides studied. Are the structure of the cited polysaccharides determined? Or just the composition? Is there is any structure/function studies? The authors list some example without objective criticism on the conducted research. Why they selected such or such examples? Because these examples are key works in the field?

I was interested to read the section 4 “Challenges for the commercialization of marine microbial polysaccharides” but I was a bit disappointed at the end. Indeed, I was expected to find some success stories dealing with marine microbial polysaccharides which reach the market. I know there are in the cosmetic industry. Also, a survey of the deposited patent on the EPS could have given an idea of the development of potential new products. The authors give some tracks to explain the low success of the marine polysaccharides to reach the market. I suggest that biomedical applications need in depth structural characterization of the products to reach the market and it is also a break, I suppose, to do in vivo experiments.

Altogether, the article needs to be improved and completed with a better literature and patent surveys to enrich the document with some illustrations and solid statements.

Author Response

Reviewer:

the authors didn’t give any element on the state of art concerning the number of new polysaccharides isolated which composition and structure have been determined the last decade (or in general). Therefore, it is difficult to appreciate what effort is involved by the scientific community on this topic.

Authors:

we have highlighted same. In addition, we have compiled data and represent them in tables to showcase extant research done.

We have also attempted to answer other queries as well and  the composition are listed in the tables shown. However we have not gone into details about the structure. This will entail other aspects where in depth analysis will be needed to document their potential functions.

For the last part (concluding remark), again, we did not go into details but did state the commercial aspect with examples (some examples have been highlighted as well above in the text). No details are required as many of these compounds have not been tested as such and very rarely clinical trials have been attempted. Hence studies are still in transitional stages.

Reviewer 2 Report

In this manuscript the authors performed a comprehensive review on several aspects related to microbial polysaccharides of marine origin, mainly on their applications and related characteristics. This topic has scientific relevance, however some information should be more detailed. Therefore, I believe that major alterations should be considered prior to publication:

1.       The authors chose to focus only on bacteria and fungi, although there are other marine microorganisms that produce polysaccharides, as indicated in Ref. 15 for example. Therefore, the authors should clearly and briefly explain in the manuscript why they decided to focus only on those microorganisms.  

2.       In general, while the concepts and theoretical parts are well explained, more details should be added to the examples that are being given. Some sentences as an example of this: Lines 101-102 name the biological activities that were detected; Lines 219-221 describe the most relevant novel species or polymers discovered; Lines 226-227 name the probiotic marine bacteria; Lines 256-258 name the microorganisms that produce the mentioned polysaccharides; Lines 324-326 give examples of polymers from marine microorganisms being already commercialized; Lines 342-353 at least one example of a relevant microorganism genetically manipulated should be given.

3.       I suggest to summarize some of the information in table(s) to be more engaging to the reader and make it easier to find the data.

4.       Lines 330-332 rephrase the sentence in order to be clear with what other organisms/polysaccharides the comparison is being made.

5.       Confirm that all strains are written in italics, which is especially missing in the references section.

English with only a few minor inaccuracies, but should be carefully revised throughout the manuscript.

Author Response

comment 1:

we focus on bacteria for various reasons (eg. ease of isolation, abundance and high diversity). In addition,most of the studies done have targeted bacteria instead of fungi. However, existing research done on fungi has been stated in the paper.

2. comments have been addressed. We thank the reviewer for his comments.

3. Yes, many thanks. We have included tables and feel that the paper if far better now.

4 & 5: yes, we have made all changes. we have double checked all scientific names in refs list and italicised them

Round 2

Reviewer 2 Report

All comments raised have been taken into consideration, therefore I recommend this manuscript for publication in its current form.  

Author Response

Thank you.
